# Prevalence of Newly Diagnosed Malignancies in Patients with Polymyalgia Rheumatica and Giant Cell Arteritis, Comparison of 18F-FDG PET/CT Scan with Chest X-ray and Abdominal Ultrasound: Data from a 40 Week Prospective, Exploratory, Single Centre Study

**DOI:** 10.3390/jcm9123940

**Published:** 2020-12-04

**Authors:** Amir Emamifar, Søren Hess, Torkell Ellingsen, Susan Due Kay, Jacob Christian Bang, Oke Gerke, Per Syrak Hansen, Ziba Ahangarani Farahani, Henrik Petersen, Niels Marcussen, Inger Marie Jensen Hansen, Peter Thye Rønn

**Affiliations:** 1Department of Clinical Research, Faculty of Health Sciences, University of Southern Denmark, 5000 Odense, Denmark; Henrik.Petersen@rsyd.dk (H.P.); Peter.Thye-Ronn@rsyd.dk (P.T.R.); 2Diagnostic Center, Svendborg Hospital, OUH, 5700 Svendborg, Denmark; Per.Syrak.Hansen@rsyd.dk; 3Department of Rheumatology, Svendborg Hospital, OUH, 5700 Svendborg, Denmark; Susan.Due.Kay@rsyd.dk (S.D.K.); imjh@carlhansen.dk (I.M.J.H.); 4OPEN, Odense Patient data Explorative Network, Odense University Hospital, 5000 Odense, Denmark; 5Department of Radiology and Nuclear Medicine, Hospital of Southwest Jutland, 6700 Esbjerg, Denmark; soren.hess@rsyd.dk; 6Institute of Regional Health Research, Faculty of Health Sciences, University of Southern Denmark, 5000 Odense, Denmark; 7Rheumatology Research Unit, Odense University Hospital and University of Southern Denmark, 5000 Odense, Denmark; torkell.ellingsen@rsyd.dk; 8Department of Radiology, Svendborg Hospital, OUH, 5700 Svendborg, Denmark; Jacob.Christian.Bang@rsyd.dk; 9Research Unit of Clinical Physiology and Nuclear Medicine, Department of Clinical Research, University of Southern Denmark, 5000 Odense, Denmark; Oke.Gerke@rsyd.dk; 10Department of Nuclear Medicine, Odense University Hospital, 5000 Odense, Denmark; ziba.farahani2@rsyd.dk; 11Department of Pathology, Odense University Hospital, 5000 Odense, Denmark; Niels.Marcussen@rsyd.dk

**Keywords:** polymyalgia rheumatica, giant cell arteritis, cancer, 18F-FDG PET/CT

## Abstract

The aim of the study was to identify the prevalence of newly diagnosed malignancies in patients with polymyalgia rheumatica (PMR) and giant cell arteritis (GCA), with the aid of 18F-FDG PET/CT scan compared to conventional imaging techniques: Chest X-ray (CXR) and abdominal ultrasound (US). Secondarily, to examine the relative diagnostic accuracy of these two imaging modalities for the detection of cancer. Eighty consecutive patients with newly diagnosed PMR, GCA, or concomitant PMR and GCA, were included and followed up for 40 weeks. All patients underwent an 18F-FDG PET/CT scan, CXR, and abdominal US at diagnosis. Imaging findings were dichotomously categorized into malignant or benign. Among 80 patients, three patients were diagnosed with seronegative rheumatoid arthritis and were excluded from the analysis. Of the remaining 77, 64 (83.1%) patients were diagnosed with pure PMR, 3 (3.9%) with pure GCA, and 10 (13.0%) with concomitant PMR and GCA. Five types of cancer that were more prevalent than the one-year prevalence of 1.2% among the background population were found in four (5.2%; 95%CI: 1.4–12.8%) patients. CXR/abdominal US could detect the solid cancer in one patient, whereas 18F-FDG PET/CT could identify all four solid cancers. Furthermore, four (5.2%; 95%CI: 1.4–12.8%) cases of monoclonal gammopathy of undetermined significance (MGUS) were found. An increase in C reactive protein (CRP) implicated an increased risk for cancer of 2.4% (OR: 1.024, 95%CI: 1.001–1.047; *p* = 0.041). 18F-FDG PET/CT can reveal occult cancers at an early stage with a high negative predictive value, and it is specifically beneficial in PMR/GCA patients with nonspecific symptoms.

## 1. Introduction

Research on the association between malignant diseases and polymyalgia rheumatica and/or giant cell arteritis (PMR/GCA) is scarce, and findings are still a matter of debate. Hypothetically, an increased prevalence of malignant diseases might be observed in PMR/GCA, partly due to immune system dysregulation of unknown aetiology and partly because a higher risk of cancer occurrence has been reported in patients with other autoimmune diseases such as rheumatoid arthritis [1]. Identifying a possible link between malignancies and PMR/GCA is an issue warranting consideration in clinical practice and could alter cancer screening practice in this population. Apart from a few case reports and case series, most previous studies have not found enhanced susceptibility to malignancies in patients with PMR/GCA compared to matched control groups [2,3,4,5,6]. However, few studies revealed a possible link between malignant diseases and PMR/GCA [7,8,9]. Accordingly, the prevalence of diagnosed malignancies might be affected by the clinical subtypes of the disease (for instance, pure PMR vs. pure GCA) and the patient population [10]. Furthermore, some specific cancers have been demonstrated to be connected to PMR/GCA—e.g., skin cancers [8]—and haematologic cancers such as acute myeloid leukaemia [11], multiple myeloma/monoclonal gammopathy of undetermined significance (MGUS) [12], and myeloproliferative neoplasm [13]. On the basis of these conflicting findings, no definitive conclusions can be drawn. However, regardless of the increased or even similar prevalence of malignancies in PMR/GCA patients, they are an important differential diagnosis because they can mimic the symptoms of PMR/GCA and should be suspected during the course of the disease.

The high diagnostic accuracy of fluorine-18-fluorodeoxyglucose positron emission tomography/computed tomography (18F-FDG PET/CT) to detect PMR and GCA has been reported previously [14]. Moreover, 18F-FDG PET/CT is a pivotal imaging modality that can reveal occult malignancies mimicking the musculoskeletal or vasculitic symptoms of PMR/GCA [15,16,17]. 18F-FDG PET/CT can also detect the large vessel involvement of PMR/GCA with high sensitivity and specificity [18,19]. Taken together, 18F-FDG PET/CT is a useful tool in the diagnostic workup of patients with PMR/GCA. Thus, to add to the current understanding of malignant comorbidities in patients with PMR/GCA, the intent of the current exploratory study was to determine the prevalence of newly diagnosed malignancies in patients with PMR and GCA, with the aid of 18F-FDG PET/CT scan compared to nationally recommended, conventional imaging techniques: Chest X-ray (CXR) and abdominal ultrasound (US). The relative diagnostic accuracy of these two imaging modalities for the detection of malignancies in PMR/GCA patients was also examined.

## 2. Materials and Methods

### 2.1. Study Design and Setting

This was a prospective, exploratory, and single center cohort study. The study was performed at the Diagnostic Center in collaboration with the Section of Rheumatology, Svendborg Hospital, Svendborg, Denmark, between February 2018 and December 2019. The CXR and abdominal US were performed at the Department of Radiology, Svendborg Hospital, and 18F-FDG PET/CT scans were undertaken at the Department of Nuclear Medicine, Odense University Hospital, Odense, Denmark. Ethical approval was granted by the Regional Ethics Committee of the Region of Southern Denmark (identification number: S-20160098) and the Danish Data Protection Agency (J.nr 16/40522). This study was also registered at ClinicalTrials.gov (Identifier: NCT02985424).

### 2.2. Participants

Eighty consecutive patients with newly suspected PMR, GCA, or concomitant PMR and GCA were included in the study after an initial visit at the Diagnostic Center. Patients with clinical suspicion of PMR/GCA were referred by general practitioners to our Diagnostic Center and subsequently followed in the general rheumatology clinic at Svendborg hospital when the initial diagnosis was confirmed. Inclusion and exclusion criteria have previously been described in detail and summarized below [20,21]. All included patients gave written informed consent.

Inclusion criteria for PMR:Age ≥50 years;Bilateral shoulder or hip pain;Morning stiffness >45 min;Elevated erythrocyte sedimentation rate (ESR) and/or C reactive protein (CRP);Disease duration >2 weeks.

Inclusion criteria for Cranial-GCA (C-GCA):Age ≥50 years;Elevated ESR and/or CRP;Scalp tenderness;Vision disturbances;Headache (new or changed);Jaw claudication;Tenderness of the temporal arteria.

For C-GCA, the cut off value of 50 years for age and elevated ESR and/or CRP must present together with at least two of other symptoms related to vasculitis. However, if concomitant PMR was also present in an individual patient, one cranial symptom was sufficient. Patients with clinical suspicion of Large Vessel-GCA (i.e., upper extremity claudication and upper extremity blood pressure discrepancies) were also eligible for inclusion.

Exclusion criteria:Infections, malignancy, or any other conditions that prednisolone was unsuitable to initiate;Contraindication to 18F-FDG PET/CT (blood glucose >145 mg/dL after 6 h fasting);Initiation of steroid treatment more than 3 days prior to 18F-FDG PET/CT;Inability to provide informed consent;Patients with dementia or inability to communicate in Danish.

Treatment with oral glucocorticoids was initiated according to national guidelines, with 20 to 30 mg/day in the case of PMR and up to 75 mg/day if GCA was suspected [20]. To ensure the diagnosis of PMR/GCA, all patients were followed up for 40 weeks after the initial diagnosis, a time interval that is considered sufficient to secure an accurate diagnosis [22,23]. Furthermore, a unilateral temporal artery biopsy (TAB) was performed at the Department of Otolaryngology at the time of diagnosis or shortly thereafter [20]. A TAB was considered positive if there was sign of active arteritis (defined as ongoing inflammation of the media) or healed arteritis (defined as fibrosis, attenuation and/or neovascularization of the media, irregular intimal proliferation, multifocal to complete loss of internal elastic lamina, adventitial fibrosis while no ongoing chronic medial inflammation exists) [24].

### 2.3. Data Collection

Patient demographics, clinical evaluation, Charlson comorbidity index score [25], laboratory tests, TAB, and findings on CXR/abdominal US and 18F-FDG PET/CT were collected and managed by means of REDcap (Research Electronic Data Capture), a secure, web based software platform designed to support data capture for research studies in the Open Patient Data Explorative Network [26].

### 2.4. Imaging Technique

CXR and abdominal US were performed according to standard procedures at the Department of Radiology before the initial visit or on the same day. Abdominal US was performed by a specialist with either a Philips EPIQ7G with a C1-5 curved array transducer or a GE LOGIQ E9 with a C1-6 curved array transducer. CXR was acquired in two projections. Additionally, all included patients underwent an 18F-FDG PET/CT scan before initiation of glucocorticoids, or within 3 days in the case of patients with a suspected diagnosis of GCA. As previously described, 18F-FDG-PET/CT data were acquired on a GE PET/CT scanner (General Electrics, Milwaukee, WI, USA) [21]. A weight-adjusted FDG dose of 4 MBq/kg (min. 200 MBq, max. 400 MBq) was injected intravenously after a fasting period of at least 6 h prior to the scan. For anatomic correlation and attenuation correction of PET images, a low dose CT without contrast enhancement was obtained from the vertex of the skull to the proximal femora. Subsequently, a PET scan of the same area was acquired with 2.5 min per bed position. Images were reconstructed using iterative reconstruction and displayed in coronal, axial, and sagittal planes.

The two applied imaging modalities were evaluated separately. The CXR and abdominal US images were evaluated as part of the daily routine by the radiologists from the Department of Radiology blinded to 18F-FDG-PET/CT findings. The 18F-FDG-PET/CT images were reviewed by two experienced nuclear medicine physicians at the Department of Nuclear Medicine blinded to clinical information other than project inclusion. Scans were assessed visually according to a standard procedure based on international guidelines, a composite gestalt interpretation based on pattern recognition of increased metabolic activity (pathologic 18F-FDG avidity) and correlated with anatomic information (e.g., morphological changes and contrast enhancement). Imaging findings in all three modalities were dichotomously categorized into “malignant” (i.e., suspicious findings requiring further diagnostic workup) or “benign” in a descriptive manner by senior readers. Accordingly, 18F-FDG-PET/CT images were interpreted based on FDG uptake intensity and pattern and by integrating the anatomic information provided by CT. Findings suggestive of cancer were evaluated by a senior consultant at the Diagnostic Center and referred to the relevant department for further diagnostic workup.

Relative PMR/GCA FDG uptake patterns have been described previously [21]. A 4 point visual grading scale (VGS) was used to score the FDG uptakes at eight articular/periarticular sites and 14 arterial segments with 0 = no uptake; 1 = slight but not negligible uptake, lower than liver uptake; 2 = intermediate uptake, equivalent to liver uptake; 3 = high grade uptake, higher than liver uptake. Afterward, the global inflammation burden—total PMR (range: 0–24) and GCA (range: 0–42) scores—was defined as the sum of VGS at each articular/periarticular site or arterial segment, considering the highest score of the bilateral sites.

### 2.5. Reference Standard

If any kind of malignant disease was detected or developed during the study, the patient was referred to the relevant departments for detailed assessment and subsequent treatment. Verification of malignant diseases in the patients was confirmed by histology, cytology, or, if necessary, by other imaging modalities.

### 2.6. Partial Patient and Public Involvement

The present study was supported by a patient advisory group that provided input to the research questions. Patients partnered with us for the design of the study, the informational material to support the intervention, and the burden of the intervention from the patient’s perspective—e.g., the number of blood samples and the interval between them. However, additional patient involvement was difficult due to the very technical nature of the methods used in the study.

### 2.7. Statistical Analysis

Data are presented as frequencies (percentages), mean ± standard deviation (SD), or median (interquartile range (IQR]) depending on data type and distribution. The prevalence of cancers in the present cohort was compared to the age, gender, and region matched one-year prevalence of all cancer sites in Denmark in 2016 [27]. A comparison of two binary variables was performed using the chi square test or Fisher’s exact test (depending on cell frequencies). The comparison of continuous variables was performed using a Student’s t test or Wilcoxon rank sum test (Mann–Whitney U test), depending on the assumption of normally distributed data. McNemar’s test was used to compare the relative diagnostic accuracy of 18F-FDG PET/CT versus CXR/abdominal US for detecting malignancies in the patients. Odds ratios (OR), estimated by logistic regression, and their respective 95% confidence intervals (95% CI) were used to measure the magnitude of the associations between cancer and the covariates age, gender, smoking status, CRP, ESR, and total PMR and GCA scores. A p value was considered significant if *p* < 0.05. No method of imputation was used for missing data. Data on the patients who were diagnosed with rheumatic diseases other than PMR/GCA at any time after inclusion were omitted from the statistical analysis. The statistical analysis plan has previously been reported in detail [21]. Statistical analysis was performed using STATA version 16.0 (StataCorp, College Station, TX, USA).

## 3. Results

Eighty consecutive patients were included in this study and followed up for 40 weeks. Three patients were diagnosed with rheumatoid arthritis during the follow-up period. Statistical analyses were performed in 77 patients. Among 77 patients, 64 (83.1%) patients were diagnosed with pure PMR, 10 (13.0%) with concomitant PMR and GCA, and 3 (3.9%) with pure GCA, and the diagnoses were confirmed during the follow-up period. The baseline characteristics of the included patients in the present cohort have previously been published and are briefly reported in Table 1 [21].

A TAB was performed in 70 patients and was positive in seven (10%) patients. Active arteritis was found in four (5.7%) and healed arteritis in three (4.3%).

### Prevalence of Malignancies

Abnormal findings on CXR/abdominal US and 18F-FDG PET/CT are listed in Table 2 and Table 3, respectively.

During the study period, five cancers were found in four (5.2%; 95% CI: 1.4–12.8%) patients and confirmed by histopathologic examination—i.e., breast cancer (*n* = 2), colon cancer (*n* = 2), and a case of skin cancer (basal cell carcinoma) in one of the patients with colon cancer. CXR/abdominal US detected the solid cancer in one patient (out of four), whereas 18F-FDG PET/CT could identify all four solid cancers. Even though 18F-FDG PET/CT could correctly detect all four solid cancers, the results were not significantly different from the findings on CXR/abdominal US (*p* = 0.25). Table 4 gives 18F-FDG PET/CT and CXR/abdominal US findings in every patient with histologically confirmed cancer.

CXR/abdominal US displayed probable malignant findings in eight patients, one case was confirmed to be malignant (one of the abovementioned breast cancers), whereas the suspicion of malignancy was dismissed upon further examination in the remaining seven patients. 18F-FDG PET/CT displayed probable malignant findings in nine patients, with four cases confirmed malignant (the abovementioned two cases of breast cancer and two cases of colon cancer), whereas the suspicion was dismissed in the remaining five out of nine cases (two benign breast lesions, oesophagitis, an unchanged lung nodule, and normal findings in the oral cavity). Thus, the false positive rates of the suspected malignant findings in 18F-FDG PET/CT and CXR/abdominal US were five out of nine (55.6%) and seven out of eight (87.5%), respectively, whereas overall false positive rates were 6.5% (5/77) and 9.1% (7/77), respectively.

Furthermore, we found four (5.2%; 95% CI: 1.4–12.8%) cases of MGUS in the included patients (Table 1). Appendix A summarized the prevalence of solid cancers and MGUS according to age groups and gender. To compare the prevalence of cancers in our cohort to the background population, the age, gender, and region matched one-year prevalence of all cancer sites in Denmark in 2016 was extracted from the NORDCAN database (Appendix A) [27].

In an exploratory data analysis of those with and without solid cancers as well as MGUS, age was the only variable that was statistically significantly higher in the patients with solid cancers (*p* = 0.049). In addition, CRP was significantly higher in patients with MGUS (*p* = 0.017). Even though all four solid cancers were detected in pure PMR patients and no cancer were found in patients with pure GCA and concomitant PMR and GCA, the result was not statically significant (*p* = 0.99). With respect to the total PMR and GCA scores, no statically significant differences were found in patients with and without solid cancers and MGUS (Table 5).

In an additional exploratory logistic regression analysis, an increase in CRP implicated, on average, an increased risk for cancer of 2.4% (OR: 1.024, 95% CI: 1.001–1.047; *p* = 0.041), when the cancer was either a solid cancer or MGUS (*n* = 8) (Table 6).

## 4. Discussion

To our knowledge, this is the first exploratory study of its kind that prospectively compared the prevalence of new malignancies in patients with PMR/GCA during 40 week follow-up with the prevalence of malignancies in a regional age and gender matched control population. A solid cancer was detected in four patients (5.2%; 95% CI: 1.4–12.8%) among 77 newly diagnosed PMR and GCA patients. Moreover, four (5.2%; 95% CI: 1.4–12.8%) patients were diagnosed with MGUS. 18F-FDG PET/CT correctly detected all four patients. Numbers of benign lesions with increased FDG uptake simulating malignant lesions led to false positive results of 18F-FDG PET/CT. Even though this is a potential pitfall in the interpretation of 18F-FDG PET/CT, the false positive rate observed with 18F-FDG PET/CT was lower than the respective rate seen with CXR/abdominal US (55.6% vs. 87.5%, and in total 6.5% vs. 9.1%) in this study. Owing to the small number of cancers among patients, results did not achieve statistical significance when the diagnostic accuracy of 18F-FDG PET/CT was compared to the conventional method of imaging—i.e., CXR/abdominal US. The global burden of inflammation in 18F-FDG PET/CT did not differ between the patients with and without solid cancers or in patients with MGUS.

The overall prevalence of solid cancers in the present cohort appears to be higher than the one-year prevalence of 1.2% among the age, gender, and region matched background population in 2016 [27]. However, the prevalence of all cancer sites in the matched background population did not even cover the confidence interval found in the present study: 1.2% vs. 95% CI: 1.4–12.8%, although the difference was slight. In fact, it was difficult to draw any definite conclusion because it would not be ethically acceptable to perform 18F-FDG PET/CT in healthy individuals and, therefore, recruitment of a formal control group was not feasible.

CRP is a classic acute phase reactant that is elevated in response to acute and chronic inflammation as well as cancer [28]. Data from the Danish general population cohort of cancer free individuals followed up for up to 16 years showed that elevated levels of CRP at baseline were associated with increased risk of cancer of any type and with some specific cancers such as lung cancer and possibly colorectal cancer [29]. In line with previous findings, our cohort demonstrated that an increase in CRP is associated with an increased risk for cancer of 2.4% in PMR/GCA patients. Whether the circulating level of CRP can play a causal role in the pathogenesis of comorbid cancers in PMR/GCA, or it is simply a marker of occult cancer in this group of patents, is a matter of interest and should be addressed in future research.

Previous research on the prevalence of malignancies in patients with PMR/GCA is limited [2,3,4,5,6,7,8,9,10,11,12,30,31]. A recent systematic review by Partington et al. on comorbidities in patients with PMR did not find strong evidence for an excess malignancy rate in their patients [32]. The authors argued that the excess rate observed following the cancer diagnosis could be due to an element of misdiagnosis because PMR and some cancers share several features—e.g., myalgia, fatigue, and weight loss—but also because number of cases of cancer decreased to that in the background population as time increased after the initial diagnosis. In another metaanalysis by Ungprasert et al., the pooled risk ratio of malignancy in the patients with PMR/GCA was equal to 1.14 (95% CI: 1.05–1.22) [33]. A higher risk of malignancy in the first 6 to 12 months after diagnosis was reported by the authors, with a pooled risk ratio of 2.16 (95%CI: 1.85–2.53). Methodologically, registry-based studies such as those of Ji et al. and Muller et al. suffer from the risk of misclassification and misdiagnosis despite large study populations [8,9]. On the other hand, results of cohort studies are often more accurate because the diagnosis of PMR/GCA is confirmed by a rheumatologist; however, the study populations are small and not comparable with those in the registry based studies [2,3,4,5,6].

The strengths of this study include the following. Firstly, this was a prospective study in which all included patients were followed up for 40 weeks. This duration of time seemed to be sufficient to ensure a certain diagnosis of PMR/GCA and minimize the risk of misdiagnosis and misclassification often seen in registry-based studies. Secondly, 80 consecutive PMR/GCA patients were included over approximately one year. Thirdly, with respect to the diagnostic accuracy of the two imaging strategies, each included patient was assessed in relation to his/her own control. The strength of paired diagnostic studies compared to randomized controlled studies has been discussed previously [34,35]. The use of paired data, as in the present study, contributed to minimizing the risk of confounding together with the need for stratification analysis. Lastly, our study population was not limited to a specific subtype of the disease but consisted of a real life PMR and GCA population with varied phenotypes of the disease such as that seen in clinical settings.

This study had some limitations. First of all, due to the fear of ischaemic complications and blindness, 18F-FDG PET/CT was performed within 3 days of treatment initiation with high-dose prednisolone (up to 75 mg/day) in patients suspected of having GCA. For this reason, glucocorticoid use might have interfered with the uptake of FDG and altered the detection rate of malignancies in the patients, even though this effect has previously been found to be negligible [36,37]. Secondly, patients were followed up for 40 weeks after the initial diagnosis of PMR/GCA. Even though a high negative predictive value of 18F-FDG PET/CT correlates with a low incidence of cancer during follow-up, this timeframe is relatively short to detect cancers that occur beyond the first year after the initial diagnosis of PMR/GCA [38]. Thirdly, the abdominal US was not performed by an individual assessor and, therefore, the robustness and homogeneity of the US results might have been compromised. Additionally, our study had limited power to produce significant results with regard to the diagnostic accuracy of the two imaging modalities used. This study was, however, the first of its kind; hence, formal power calculations were impossible to conduct due to the lack of a previous comparable study in the literature and the absence of experience in the local clinical daily routine. Finally, it is worth considering that the prevalence of cancer in PMR/GCA in the present cohort might be underestimated since cancer screening in PMR/GCA in the daily clinical routine is mostly limited to the patients with atypical presentation and insufficient glucocorticoid response, where the chance of cancer is probably higher.

The results of the present study have a high degree of generalizability because the patients studied were from a general rheumatology clinic at a regional hospital, representing the real-life PMR and GCA population with varied phenotypes of the disease. The generalizability of our study, however, might be limited because it was performed at a single institution. In addition, the cost of 18F-FDG PET/CT and its availability may hamper the generalizability of this method in some international centers, although it is largely accessible in Denmark and expenses are covered by the Danish health insurance system.

## 5. Conclusions

In conclusion, we found a higher prevalence of cancer in patients with PMR/GCA during a 40 week follow-up compared to a gender, age, and region matched background population. However, because of the exploratory design of this study with no matched control group, our results should be confirmed by future studies. Eliminating a diagnosis of malignant disease in patients with PMR/GCA is important, especially in those with atypical symptoms and poor response to glucocorticoids. 18F-FDG PET/CT can reveal occult cancers at an early stage with fair precision, and perhaps more importantly, can rule out cancer with high negative predictive value in patients with nonspecific symptoms suggestive of malignancies. Thus, the implication of our diagnostic setup emphasizes the need for 18F-FDG PET/CT, or at least a diagnostic CT scan, together with a broad biochemical package including measurement of M component in the diagnostic workup of PMR/GCA patients for an accurate and early diagnose of occult cancer.

## Figures and Tables

**Table 1 jcm-09-03940-t001:** Characteristic of the included patients (*n* = 77).

Age, y	71.8 ± 8.0
Gender, *n* (%)	
Female	49 (63.6%)
Body mass index, kg/m^2^; median (IQR)	25.4 (22.1–27.9)
Smoking status, *n* (%)	
Never smoker	25 (32.5%)
Smoker (including former smoker)	52 (67.5%)
Alcohol status, *n* (%)	
≤6 units per week	56 (72.7%)
>6 units per week	21 (27.3%)
Duration of symptoms before diagnosis, *n* (%)	
2 to 4 weeks	18 (23.4%)
5 to 8 weeks	32 (41.6%)
9 to 16 weeks	16 (20.9%)
17 to 24 weeks	4 (5.2%)
>24 weeks	7 (9.1%)
Initial symptoms, *n* (%)	
Constitutional symptoms, *n* (%)	74 (96.1%)
Shoulder girdle symptoms, *n* (%)	72 (93.5%)
Hip girdle symptoms, *n* (%)	68 (88.3%)
Cranial symptoms, *n* (%)	19 (24.7%)
Charlson comorbidity index score; median (IQR)	3 (2–4)
Lab results, (reference values)	
Hemoglobin, mmol/L (8.3–10.5); median (IQR)	7.6 (7.2–8.1)
Leucocytes, 1 × 10^9^ /L (3.50–8.80); median (IQR)	9.39 (8.05–11.2)
Platelet, 1 × 10^9^ /L (145–350); median (IQR)	348 (298–451)
ESR, mm (2–20); median (IQR)	54 (38–77.5)
CRP, mg/L (<6.0); median (IQR)	37 (17–64)
Fibrinogen, µmol/L (5.2–12.6); median (IQR)	14.9 (13–17.5)
Prostate specific antigen, µg/L (<5.0); median (IQR)	2.35 (0.95–7.2)
M component positive, *n* (%)	4 (5.2%)

ESR: Erythrocyte sedimentation rate; CRP: C reactive protein.

**Table 2 jcm-09-03940-t002:** Chest X-ray (CXR)/Abdominal ultrasound (US) abnormal findings.

Probably Malignant Findings Requiring Further Workup, *n* (%)	
Gallbladder	5 (6.5%)
Breast	1 (1.3%)
Lung	1 (1.3%)
Bladder	1 (1.3%)
Benign findings, *n* (%)	
Renal (cyst, angiomyolipoma)	25 (32.5%)
Sings of obstructive pulmonary disease	22 (28.6%)
Fatty liver	15 (19.5%)
Liver cyst	11 (14.3%)
Cholecystolithiasis	10 (13.0%)
Chronic nephropathy	6 (7.8%)
Prostate hypertrophy	5 (6.5%)
Lung infiltration	5 (6.5%)
Non-specific	4 (5.2%)
Pleuritis	3 (3.9%)
Pancreatic cyst	1 (1.3%)
Uterus fibroma	1 (1.3%)
Atrophic pancreas + calcification	1 (1.3%)
Aortic aneurysm	1 (1.3%)

**Table 3 jcm-09-03940-t003:** 18F-FDG PET/CT abnormal findings.

FDG Uptakes, Probable Malignant Findings Requiring Further Workup, *n* (%)	
Breast	4 (5.2%)
Gastrointestinal tract	3 (1.3%)
Oral cavity	1 (1.3%)
Lung	1 (1.3%)
FDG uptakes, benign findings, *n* (%)	
Lower gastrointestinal tract	10 (13.0%)
Lymph nodes	10 (13.0%)
Thyroid	5 (6.5%)
Tonsil	5 (6.5%)
Costae	4 (5.2%)
Parotid	2 (2.6%)
Testis	2 (2.6%)
Neck	1 (1.3%)
Tongue	1 (1.3%)
Adrenal	1 (1.3%)
Heart	1 (1.3%)
Kidney	1 (1.3%)
Abdominal wall	1 (1.3%)
Frontal sinus	1 (1.3%)
Gluteal muscle	1 (1.3%)
Pacemaker infection	1 (1.3%)

**Table 4 jcm-09-03940-t004:** 18F-FDG PET/CT cancer findings compared to CXR and abdominal US in patients with confirmed cancer by histology together with the sum of visual grading scale (VGS) in articular/periarticular and arterial segments.

Cancer Type in the Patients (*n* = 4)	18F-FDG PET/CT findings	Time of Cancer Diagnosis after Initial Diagnosis of PMR/GCA	CXR/Abdominal US	Total PMR Score	Total GCA Score
Colon adenocarcinoma + Basal Cell Carcinoma	colon (malignant), thyroid (benign)	<1 month *	chronic obstructive changes/renal cyst, liver cyst	0	1
Colon adenocarcinoma	colon (malignant), testis (benign)	<1 month	Emphysema, unspecific lung changes/renal cyst, liver cyst	10	0
Breast cancer	Breast (malignant), colon (benign)	<1 month	Emphysema, calcification in breast/normal	16	0
Breast cancer	Breast (malignant)	<1 month	Normal	14	0

* Cancer was suspected in less than one month of PMR/GCA diagnosis, but it is confirmed later as the patient was initially not interested in further diagnostic work up. PMR/GCA: polymyalgia rheumatica and/or giant cell arteritis.

**Table 5 jcm-09-03940-t005:** Exploratory analysis of data in patients with and without solid cancer as well as monoclonal gammopathy of undetermined significance (MGUS).

Variables	− Cancer, *n* = 73	+ Cancer, *n* = 4	*p* Value	− MGUS, *n* = 73	+ MGUS, *n* = 4	*p* Value
Age, mean ± SD	71.4 ± 7.8	79.7 ± 7.5	0.049 ^1^	71.9 ± 8.0	70.2 ± 9.2	0.79 ^1^
Gender, *n* (%)						
Female	46 (59.7%)	3 (3.9%)	0.99 ^2^	47 (61.0%)	2 (2.6%)	0.62 ^2^
Body mass index; median (IQR)	25.4 (22.5–27.9)	23.7 (21.7–25.6)	0.40 ^1^	25.3 (22–27.9)	26.2 (25.2–26.9)	0.57 ^1^
Smoking, *n* (%)			0.99 ^2^			0.30 ^2^
Never smoker	24 (31.2%)	1 (1.3%)		25 (32.5%)	0 (0%)	
Smoker (including former smoker)	49 (63.6%)	3 (3.9%)		48 (62.3%)	4 (5.2%)	
Alcohol, *n* (%)			0.30 ^2^			0.99 ^2^
≤6 units per week	54 (70.1%)	2 (2.6%)		53 (68.8%)	3 (3.9%)	
>6 units per week	19 (24.7%)	2 (2.6%)		20 (26.0%)	1 (1.3%)	
Duration of symptoms, *n* (%)			0.56 ^2^			0.99 ^2^
2 to 4 weeks	17 (22.1%)	1 (1.3%)		17 (22.1%)	1 (1.3%)	
5 to 8 weeks	30 (39.0%)	2 (2.6%)		30 (39.0%)	2 (2.6%)	
9 to16 weeks	16 (20.8%)	0 (0%)		15 (19.5%)	1 (1.3%)	
17 to 24 weeks	4 (5.2%)	0 (0%)		4 (5.2%)	0 (0%)	
>24 weeks	6 (7.8%)	1 (1.3%)		7 (9.1%)	0 (0%)	
Constitutional symptoms, *n* (%)	70 (90.9%)	4 (5.2%)	0.99 ^2^	70 (90.9%)	4 (5.2%)	0.99 ^2^
Shoulder girdle symptoms, *n* (%)	68 (88.3%)	4 (5.2%)	0.99 ^2^	68 (88.3%)	4 (5.2%)	0.99 ^2^
Hip girdle symptoms, *n* (%)	65 (84.4%)	3 (3.9%)	0.40 ^2^	64 (83.1%)	4 (5.2%)	0.99 ^2^
Cranial symptoms, *n* (%)	19 (24.7%)	0 (0%)	0.57 ^2^	17 (22.1%)	2 (2.6%)	0.25 ^2^
Patients pain VAS score; median (IQR)	75 (50–85)	62.5 (50–75)	0.53 ^1^	72.5 (50–80)	87.5 (77.5–95)	0.07 ^1^
Patients global VAS score; median (IQR)	80 (60–90)	62.5 (50–75)	0.37 ^1^	80 (60–90)	89.5 (79.5–95)	0.23 ^1^
Physician global VAS score; median (IQR)	30 (25–40)	24.5 (20–29)	0.15 ^1^	30 (22.5–40)	37.5 (32.5–45)	0.17 ^1^
Charlson comorbidity index; median (IQR)	3 (2–4)	4.5 (3.5–5)	0.10 ^1^	3 (2–4)	3.5 (2–4)	0.79 ^1^
Hemoglobin, mmol/L (8.3–10.5); median (IQR)	7.6 (7.2–8.2)	7.2 (6.9–7.7)	0.31 ^1^	7.5 (7.2–8)	8.3 (7.7–8.7)	0.15 ^1^
ESR, mm (2–20); median (IQR)	53.5 (38–77.5)	62.5 (37.5–76)	0.98 ^1^	54 (38–77.5)	57.5 (39–73.5)	0.99 ^1^
CRP, mg/L (<6.0); median (IQR)	37 (17–64)	34 (17.0–76)	0.80 ^1^	33 (17–60)	98 (68–115)	0.017 ^1^
Fibrinogen, µmol/L (5.2–12.6); median (IQR)	14.9 (13.1–17.2)	15.2 (12.2–19.9)	0.90 ^1^	14.9 (13–17.2)	18 (15.9–19.8)	0.11 ^1^
TAB positive, *n* (%)	7 (10.6%)	0 (0%)	0.99 ^2^	7 (10.6%)	0 (0%)	0.99 ^2^
Total PMR score; median (IQR)	14 (10–17)	12 (5–15)	0.39 ^1^	14 (10–17)	13 (6–15.5)	0.64 ^1^
Total GCA score; median (IQR)	0 (0–0)	0 (0–0.5)	0.89 ^1^	0 (0–0)	0 (0–0)	0.34 ^1^
Clinical diagnosis, *n* (%)			0.99 ^2^			0.53 ^2^
Pure PMR	60 (77.9%)	4 (5.2%)		61 (79.2%)	3 (3.9%)	
Pure GCA	3 (3.9%)	0 (0%)		3 (3.9%)	0 (0%)	
Concomitant PMR and GCA	10 (13.0%)	0 (0%)		9 (11.7%)	1 (1.3%)	

MGUS: Monoclonal Gammopathy of Undetermined Significance, VAS: Visual analogue scale, ESR: Erythrocyte sedimentation rate, CRP: C reactive protein, TAB: Temporal artery biopsy, PMR: Polymyalgia rheumatica, GCA: Giant cell arteritis. ^1^ Wilcoxon rank-sum test. ^2^ Fisher’s exact test.

**Table 6 jcm-09-03940-t006:** Results of exploratory logistic regression considering cancer as either solid cancer or MGUS or both (*n* = 8).

Variable	Odds Ratio	95% CI	*p* Value
Age	1.090	0.977–1.217	0.12
Female gender	0.668	0.118–3.769	0.65
Smoking (including former smoker)	2.842	0.274–29.518	0.38
CRP	1.024	1.001–1.047	0.041
ESR	0.974	0.928–1.022	0.28
Total PMR score	0.874	0.754–1.013	0.07
Total GCA score	0.618	0.187–2.041	0.43

CRP: C reactive protein, ESR: Erythrocyte sedimentation rate, PMR: Polymyalgia rheumatica, GCA: Giant cell arteritis.

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
