# Peer review of "Prevalence of Newly Diagnosed Malignancies in Patients with Polymyalgia Rheumatica and Giant Cell Arteritis, Comparison of 18F-FDG PET/CT Scan with Chest X-ray and Abdominal Ultrasound: Data from a 40 Week Prospective, Exploratory, Single Centre Study"

_jcm, 2020, doi:10.3390/jcm9123940_

Round 1
Reviewer 1 Report
This manuscript from Emamifar et al. describes results of a prospective, exploratory study in patients with giant cell arteritis (GCA) and polymyalgia (PMR) with the aim to investigate the prevalence of an underlying malignancy. This was done by PET/CT and compared with results from clinical investigation, other imaging modalities and histology. The study focuses on a relevant clinical issue that applies to some patients, especially those with PMR who not respond adequately to treatment. And this is one of the main concerns that I have with the concept and the consequences of the results from the study. In the clinical context we would not screen all patients with PMR or GCA or both conditions for an underlying malignancy. Usually, only those who not respond to treatment will be screened. This means that the population from the study population is different from individuals at risk for malignancies which we see in the clinic. This should be addressed in the discussion.
Minor points
- Inclusion and exclusion criteria for the study needs to be described instead off a cross reference to another study
- What was the reason for a temporal artery biopsy in patients with PMR without clinical signs of GCA?
- What was the definition of “malignant” findings in imaging (line 134)?
- Was the diagnosis of GCA based on imaging or histology or both?
- How was the diagnosis of GCA or PMR confirmed during follow-up (line 177)?
- Table 1. Some results in the table seems to without reference to the study, eg. calcium, ALT, bilirubin, TSH and other. Please shorten the table.
- What does “healed arteritis” in TAB means? Please clarify the criteria for a positive TAB.
- Please clarify the definition of probable malignant findings in PET/CT.
- There is a high rate of “false positive” findings in imaging (55,6%). This needs to be discussed.
- PET/CT has a significant higher amount of radiation compared with other imaging modalities. This needs to be discussed if this modality is considered as the first screening tool in selected patients.
Author Response
Dear reviewer and editor,
Thank you for your insightful comments. Please find our responses below.
Reviewer 1
This manuscript from Emamifar et al. describes results of a prospective, exploratory study in patients with giant cell arteritis (GCA) and polymyalgia (PMR) with the aim to investigate the prevalence of an underlying malignancy. This was done by PET/CT and compared with results from clinical investigation, other imaging modalities and histology. The study focuses on a relevant clinical issue that applies to some patients, especially those with PMR who not respond adequately to treatment. And this is one of the main concerns that I have with the concept and the consequences of the results from the study. In the clinical context we would not screen all patients with PMR or GCA or both conditions for an underlying malignancy. Usually, only those who not respond to treatment will be screened. This means that the population from the study population is different from individuals at risk for malignancies which we see in the clinic. This should be addressed in the discussion.
We added the following to the discussion part.
At last, it is worth considering that the prevalence of cancer in PMR/GCA in the present cohort might be underestimated since cancer screening in PMR/GCA in the daily clinical routine is mostly limited to the patients with atypical presentation and insufficient glucocorticoid response, where the chance of cancer is probably higher.
Minor points
- Inclusion and exclusion criteria for the study needs to be described instead off a cross reference to another study
Inclusion and exclusion are now added.
- What was the reason for a temporal artery biopsy in patients with PMR without clinical signs of GCA?
To define the agreement between TAB and PET/CT as described in the below article.
Emamifar A, et al. The Utility of 18F‐FDG PET/CT in Patients With Clinical Suspicion of Polymyalgia Rheumatica and Giant Cell Arteritis: A Prospective, Observational, and Cross‐sectional Study. ACR Open Rheumatol. 2020 Jul 22;2(8):478–90. doi: 10.1002/acr2.11163.
- What was the definition of “malignant” findings in imaging (line 134)?
We revised the text as mentioned below.
Imaging findings in all three modalities were dichotomously categorized into “malignant” (i.e. suspicious findings requiring further diagnostic workup) or “benign” in a descriptive manner by senior readers. Accordingly, 18F-FDG-PET/CT images were interpreted based on FDG uptakes’ intensity and pattern and by integrating the anatomic information provided by CT.
- Was the diagnosis of GCA based on imaging or histology or both?
It was based on the inclusion criteria (clinical symptoms) and histology.
- How was the diagnosis of GCA or PMR confirmed during follow-up (line 177)?
It is confirmed by senior rheumatologist according to the clinical symptoms, where no alternative diagnosis could better explain patients’ symptoms.
- Table 1. Some results in the table seems to without reference to the study, eg. calcium, ALT, bilirubin, TSH and other. Please shorten the table.
Table 1 is now revised.
- What does “healed arteritis” in TAB means? Please clarify the criteria for a positive TAB.
The following section has been added to the manuscript.
A TAB was considered positive if there was sign of active arteritis (defined as ongoing inflammation of the media) or healed arteritis (defined as fibrosis, attenuation and/or neovascularization of the media, irregular intimal proliferation, multifocal to complete loss of internal elastic lamina, adventitial fibrosis while no ongoing chronic medial inflammation exists).
- Please clarify the definition of probable malignant findings in PET/CT.
We revised the text as mentioned below.
All imaging findings in all three modalities were dichotomously categorized into “malignant” (i.e. suspicious findings requiring further diagnostic workup) or “benign” in a descriptive manner by senior readers. Accordingly, 18F-FDG-PET/CT images were interpreted based on FDG uptakes’ intensity and pattern and by integrating the anatomic information provided by CT.
- There is a high rate of “false positive” findings in imaging (55,6%). This needs to be discussed.
We added the following section under discussion.
- Numbers of benign lesions with increased FDG uptake simulating malignant lesions, led to false positive results of 18F-FDG PET/CT. Even though, this is a potential pitfall in the interpretation of 18F-FDG PET/CT, the false positive rate observed with 18F-FDG PET/CT was fewer than the respective rate seen with CXR/abdominal US (55.6% vs 87.5%, and in total 6.5% vs 9.1%) in this study.
- PET/CT has a significant higher amount of radiation compared with other imaging modalities. This needs to be discussed if this modality is considered as the first screening tool in selected patients.
Thanks for your comment. Although PET/CT is a powerful method for detecting cancer, its utility is currently limited by several factors for instance availability and cost, and is recommended in patients with atypical presentation who respond to glucocorticoids insufficiently when there is a suspicion of cancer.

Author Response
Dear reviewer and editor,
Thank you for your insightful comments please see our response below.
Comments to the Authors: Manuscript ID: jcm-994664
“Prevalence of newly diagnosed malignancies in patients with polymyalgia rheumatica and giant cell arteritis, comparison of 18F-FDG PET/CT scan with chest X ray and abdominal ultrasound: Data from a 40 week prospective, exploratory, single centre study”.
The authors presented the prospective research in patients with polymyalgia rheumatic and giant cell arthritis, and found a higher prevalence of cancer in patients with PMR/GCA during a 40 week follow-up compared to a background population. In conclusion they reported, that 18F-FDG PET/CT can reveal occult cancers at an early stage and is superior compared to conventional imaging techniques: chest X ray (CXR) and abdominal ultrasound (US).
The topic is very interesting, important and actual.
However, I have minor comments for the authors:
- Conventional imaging methods used in the study (CXR, abdominal US) are not appropriate methods to diagnose some solid cancers (for example colon, breast cancers) or MGUS. It is difficult to compare the diagnostic power of CXR or abdominal US with PET/CT; actually it is obvious, that PET/CT is superior. In the group of older patients, observed in the study, with nonspecific symptoms (e.g. myalgia, fatigue, weight loss, increased inflammatory parameters) cancer should be suspected. The first diagnostic step should be careful physical examination, and then e.g. breast US, endoscopy etc. PET/CT is a good diagnostic method, however it is expensive and not easily available (at least in some countries and regions). In my opinion it should be mentioned in the discussion
Conventional method (CXR and abdominal US) was recommended by the Danish rheumatism association as screening tools for cancer in PMR and GCA at that point of time when this study was designed. Although it might seem that PET/CT has superiority, this is not studied before, and as mentioned in the text we statically did not find any difference between PET/CT and conventional methods.
We also add the following to the discussion “the cost of 18F‐FDG PET/CT and its availability may hamper the generalizability of this method in some international centers, despite it is largely accessible in Denmark and expenses are covered by the Danish health insurance system.”
- Line 183: please delete “e”
Corrected.
- Line 190: please delete “found”
Corrected.
Sincerely,

Reviewer 3 Report
Dear Authors,
than You so much for Your manuscript.
I read it with attention and interest.
Here are my comments and suggestions:
- You considered PMR and GCA as an unique disease, whereas it is a common knowledge that they are two different diseases, even if sometimes associated and/or overlapped. The association between malignant diseases and PMR is different from the association with GCA. My personal opinion is that Your introduction should be re-written because is misleading in its present form.
- You wrote that "malignancies in PMR/GCA patients....should be excluded at any time in the course oif the disease" (lines-72-74). This statement is not acceptable. For an instance, if PMR or GCA are a paraneoplastic syndrome, time is commonly considered not to exceed two years. And again, "all patients were followed up for 40 weeks after the initial diagnosis, a time interval that is considered sufficient to secure an accurate diagnosis" (lines 103-104): there is a clear contradiction with what You wrote before, do not You think ?
- You wrote "Research on the association between malignant diseases and PMR/GCA is scarce....."(lines 58-59). I think that your references were incomplete. Please, consider other bibliographic entries You can find in published literature (and not taken into account. Why ?).
- During the study period, You found five types of cancer in four PMR patients. It was not clear to me how long after PMR diagnosis these cancers were diagnosed.
- The role of CRP should be better discussed. Indeed, as Calabrese et al. reported, in neoplastic patients developing PMR (or PMR-like syndromes ?) after therapy with immune-check point inhibitors, CRP serum concentrations were often in their normal range even if cancer was not in remission. Moreover, CRP serum concentrations are raised in a multitude of conditions.
Author Response
Dear reviewer and editor,
Thank you for your insightful comments please find our response below.
Dear Authors,
than You so much for Your manuscript.
I read it with attention and interest.
Here are my comments and suggestions:
- You considered PMR and GCA as an unique disease, whereas it is a common knowledge that they are two different diseases, even if sometimes associated and/or overlapped. The association between malignant diseases and PMR is different from the association with GCA. My personal opinion is that Your introduction should be re-written because is misleading in its present form.
Dear sir, thank you for your comment. In this cohort study, a population of patients with both PMR and GCA were included, with respect to several objectives described in the protocol. (Emamifar A, et al. Polymyalgia rheumatica and giant cell arteritis-three challenges-consequences of the vasculitis process, osteoporosis, and malignancy: A prospective cohort study protocol. Medicine (Baltimore). 2017 Jun;96(26):e7297. doi: 10.1097/MD.0000000000007297.) Our study was limited by few numbers of GCA where making furthere analysis ineffectual. Besides, there is no concesus respecting the nature of PMR and GCA, as several authors believe that PMR and GCA belong to the same disease entity, where PMR is at one side of disease spectrum and GCA is at the other end of the disease spectrum. Imaging findings also support this model.
Rehak Z, et al. 18F-FDG PET/CT in polymyalgia rheumatica-a pictorial review. Br J Radiol. 2017 Aug;90(1076):20170198. doi: 10.1259/bjr.20170198. Epub 2017 Jun 16.
- You wrote that "malignancies in PMR/GCA patients....should be excluded at any time in the course of the disease" (lines-72-74). This statement is not acceptable. For an instance, if PMR or GCA are a paraneoplastic syndrome, time is commonly considered not to exceed two years. And again, "all patients were followed up for 40 weeks after the initial diagnosis, a time interval that is considered sufficient to secure an accurate diagnosis" (lines 103-104): there is a clear contradiction with what You wrote before, do not You think ?
We revised this sentence:
“However, regardless of the increased or even similar prevalence of malignancies in PMR/GCA patients, they are an important differential diagnosis because they can mimic the symptoms of PMR/GCA and should be suspected during the course of the disease.”
All patients were followed for 40 weeks. This time interval is sufficient for diagnosis of PMR and GCA as described in previous leading studies. However it may not be sufficient for cancer diagnosis as we previously discussed under discussion section.
“Even though a high negative predictive value of 18F-FDG PET/CT correlates with a low incidence of cancer during follow-up, this timeframe is relatively short to detect cancers that occur beyond the first year after the initial diagnosis of PMR/GCA [34].”
- You wrote "Research on the association between malignant diseases and PMR/GCA is scarce....."(lines 58-59). I think that your references were incomplete. Please, consider other bibliographic entries You can find in published literature (and not taken into account. Why ?).
Thanks for your comment. We have almost had most relevant articles except from review and case reports articles. Howevere, we updated our reference list and added new references.
- During the study period, You found five types of cancer in four PMR patients. It was not clear to me how long after PMR diagnosis these cancers were diagnosed.
We added time of cancer diagnosis to table 4.
|
Table 4: 18F-FDG PET/CT cancer findings compared to CXR and abdominal US in patients with confirmed cancer by histology together with the sum of VGS in articular/periarticular and arterial segments. |
|||||
|
Cancer type in the patients (n = 4) |
18F-FDG PET/CT findings |
Time of cancer diagnosis after initial diagnosis of PMR/GCA |
CXR/Abdominal US |
Total PMR score |
Total GCA score |
|
Colon adenocarcinoma + Basal Cell Carcinoma |
colon (malignant), thyroid (benign) |
< 1 month* |
chronic obstructive changes/renal cyst, liver cyst |
0 |
1 |
|
Colon adenocarcinoma |
colon (malignant), testis (benign) |
< 1 month |
Emphysema, unspecific lung changes/ renal cyst, liver cyst |
10 |
0 |
|
Breast cancer |
Breast (malignant), colon (benign) |
< 1 month |
Emphysema, calcification in breast/ normal |
16 |
0 |
|
Breast cancer |
Breast (malignant) |
< 1 month |
Normal |
14 |
0 |
|
* Cancer was suspected less than one month of PMR/GCA diagnosis, but it is confirmed later as the patient was initially not interested in further diagnostic work up. |
|||||
- The role of CRP should be better discussed. Indeed, as Calabrese et al. reported, in neoplastic patients developing PMR (or PMR-like syndromes ?) after therapy with immune-check point inhibitors, CRP serum concentrations were often in their normal range even if cancer was not in remission. Moreover, CRP serum concentrations are raised in a multitude of conditions.
Thanks for your comment. PMR-like syndrome following immune check point inhibitors is considereded as immune related adverse events and is different from our patient population. Therefor results of such studies can not be compared to results. However, as we previously mentioned in the manuscipt, the causal role of CRP in the pathogenesis of comorbid cancers in PMR/GCA, should be further researched.
Dear reviewer and editor,
Thank you for your insightful comments please find our response below.
Dear Authors,
than You so much for Your manuscript.
I read it with attention and interest.
Here are my comments and suggestions:
- You considered PMR and GCA as an unique disease, whereas it is a common knowledge that they are two different diseases, even if sometimes associated and/or overlapped. The association between malignant diseases and PMR is different from the association with GCA. My personal opinion is that Your introduction should be re-written because is misleading in its present form.
Dear sir, thank you for your comment. In this cohort study, a population of patients with both PMR and GCA were included, with respect to several objectives described in the protocol. (Emamifar A, et al. Polymyalgia rheumatica and giant cell arteritis-three challenges-consequences of the vasculitis process, osteoporosis, and malignancy: A prospective cohort study protocol. Medicine (Baltimore). 2017 Jun;96(26):e7297. doi: 10.1097/MD.0000000000007297.) Our study was limited by few numbers of GCA where making furthere analysis ineffectual. Besides, there is no concesus respecting the nature of PMR and GCA, as several authors believe that PMR and GCA belong to the same disease entity, where PMR is at one side of disease spectrum and GCA is at the other end of the disease spectrum. Imaging findings also support this model.
Rehak Z, et al. 18F-FDG PET/CT in polymyalgia rheumatica-a pictorial review. Br J Radiol. 2017 Aug;90(1076):20170198. doi: 10.1259/bjr.20170198. Epub 2017 Jun 16.
- You wrote that "malignancies in PMR/GCA patients....should be excluded at any time in the course of the disease" (lines-72-74). This statement is not acceptable. For an instance, if PMR or GCA are a paraneoplastic syndrome, time is commonly considered not to exceed two years. And again, "all patients were followed up for 40 weeks after the initial diagnosis, a time interval that is considered sufficient to secure an accurate diagnosis" (lines 103-104): there is a clear contradiction with what You wrote before, do not You think ?
We revised this sentence:
“However, regardless of the increased or even similar prevalence of malignancies in PMR/GCA patients, they are an important differential diagnosis because they can mimic the symptoms of PMR/GCA and should be suspected during the course of the disease.”
All patients were followed for 40 weeks. This time interval is sufficient for diagnosis of PMR and GCA as described in previous leading studies. However it may not be sufficient for cancer diagnosis as we previously discussed under discussion section.
“Even though a high negative predictive value of 18F-FDG PET/CT correlates with a low incidence of cancer during follow-up, this timeframe is relatively short to detect cancers that occur beyond the first year after the initial diagnosis of PMR/GCA [34].”
- You wrote "Research on the association between malignant diseases and PMR/GCA is scarce....."(lines 58-59). I think that your references were incomplete. Please, consider other bibliographic entries You can find in published literature (and not taken into account. Why ?).
Thanks for your comment. We have almost had most relevant articles except from review and case reports articles. Howevere, we updated our reference list and added new references.
- During the study period, You found five types of cancer in four PMR patients. It was not clear to me how long after PMR diagnosis these cancers were diagnosed.
We added time of cancer diagnosis to table 4.
|
Table 4: 18F-FDG PET/CT cancer findings compared to CXR and abdominal US in patients with confirmed cancer by histology together with the sum of VGS in articular/periarticular and arterial segments. |
|||||
|
Cancer type in the patients (n = 4) |
18F-FDG PET/CT findings |
Time of cancer diagnosis after initial diagnosis of PMR/GCA |
CXR/Abdominal US |
Total PMR score |
Total GCA score |
|
Colon adenocarcinoma + Basal Cell Carcinoma |
colon (malignant), thyroid (benign) |
< 1 month* |
chronic obstructive changes/renal cyst, liver cyst |
0 |
1 |
|
Colon adenocarcinoma |
colon (malignant), testis (benign) |
< 1 month |
Emphysema, unspecific lung changes/ renal cyst, liver cyst |
10 |
0 |
|
Breast cancer |
Breast (malignant), colon (benign) |
< 1 month |
Emphysema, calcification in breast/ normal |
16 |
0 |
|
Breast cancer |
Breast (malignant) |
< 1 month |
Normal |
14 |
0 |
|
* Cancer was suspected less than one month of PMR/GCA diagnosis, but it is confirmed later as the patient was initially not interested in further diagnostic work up. |
|||||
- The role of CRP should be better discussed. Indeed, as Calabrese et al. reported, in neoplastic patients developing PMR (or PMR-like syndromes ?) after therapy with immune-check point inhibitors, CRP serum concentrations were often in their normal range even if cancer was not in remission. Moreover, CRP serum concentrations are raised in a multitude of conditions.
Thanks for your comment. PMR-like syndrome following immune check point inhibitors is considereded as immune related adverse events and is different from our patient population. Therefor results of such studies can not be compared to results. However, as we previously mentioned in the manuscipt, the causal role of CRP in the pathogenesis of comorbid cancers in PMR/GCA, should be further researched.

Round 2
Reviewer 3 Report
Dear Authors,
I read with attention the newer, revised version of Your manuscript.
I found that most of my comments and suggestions were satisfactorily met. Thank You.
My personal opinion is that quality of presentation and scientific soundness improved after the overall review process.
However, some points are still to be clarified.
1) You are right when you underlined that some investigators believe that PMR and GCA belong to the same disease entity. Indeed, there is the Horton's disease. The point is this: the prevalence of newly diagnosed malignancies in patients affected with "isolated" PMR is different from that we can found in patients with "isolated" GCA. Besides, CRP concentrations can be in their normal ranges in patients with GCA, while this possibility is still debated in patients with "isolated" PMR.
My personal opinion is that Your introduction section should be re-written in order to clarified these points, and that considering PMR/GCA as a whole could distort the final results of your research.
Obviously, the Academic Editor and the Editor in chief will consider whether this my suggestion and request for clarification can further improve the quality of the manuscript.
2) You updated Your references and added two new references. Please, consider to add "Muller et al. Reumatismo 2018. PMID 29589400". I thank You so much.
